processes

Article

# Gas/Liquid Operations in the Taylor-Couette Disc Contactor: Continuous Chemisorption of $CO_2$

**Georg Rudelstorfer *, Rafaela Greil, Max Vogi, Matthäus Siebenhofer, Susanne Lux and Annika Grafschafter**

Institute of Chemical Engineering and Environmental Technology, Graz University of Technology, Inffeldgasse 25C, 8010 Graz, Austria; rafaela.greil@tugraz.at (R.G.); vogi@tugraz.at (M.V.); m.siebenhofer@tugraz.at (M.S.); susanne.lux@tugraz.at (S.L.); a.grafschafter@tugraz.at (A.G.)

* Correspondence: rudelstorfer@tugraz.at; Tel.: +43-676-914-4462

**Abstract:** Gas/liquid contactors are widely used in chemical and biotechnological applications. The selection and design of bubble-column-type gas/liquid contactors requires knowledge about the gas distributor design to provide appropriate gas flow patterns. This study presents the continuous chemisorption of $CO_2$ in 0.1 molar sodium hydroxide solution in a counter currently operated gas/liquid Taylor-Couette disc contactor (TCDC). This vertical-column-type contactor is a multi-stage agitated gas/liquid contactor. The performance of a lab-size TCDC contactor in gas/liquid mass transfer operations was investigated. The apparatus design was adjusted for gas/liquid operations by installing perforated rotor discs to provide a rotational-speed-dependent dispersed gas phase holdup in the column. The parameters of dispersed gas phase holdup, volumetric mass transfer coefficient and residence time distribution were measured. In the first step, hydraulic characterization was performed. Then, the efficiency in gas/liquid operations was investigated by continuous neutralization of 0.1 molar sodium hydroxide with a gas mixture of 30 vol% $CO_2$ and 70 vol% $N_2$. Temperature, rotational speed and gas flow rate were varied. The desired pH value of pH 9 at the column outlet was kept constant by adjusting the sodium hydroxide feed. From the experimental results, the volume-based liquid-side mass transfer coefficient $k_La$ was deduced in order to model the reaction according to the two-film theory over the column height. The CSTR cascade model fitted the experimental data best. The experimental results confirm stable and efficient reactive gas/liquid contact in the Taylor-Couette disc contactor.

**Keywords:** Taylor-Couette flow; gas/liquid; chemisorption; multiphase; continuous process

**Citation:** Rudelstorfer, G.; Greil, R.; Vogi, M.; Siebenhofer, M.; Lux, S.; Grafschafter, A. Gas/Liquid Operations in the Taylor-Couette Disc Contactor: Continuous Chemisorption of $CO_2$. *Processes* **2023**, *11*, 1614. https://doi.org/10.3390/pr11061614

Academic Editors: Erik von Harbou, Michael Schlüter and Maike Baltussen

Received: 27 April 2023
Revised: 12 May 2023
Accepted: 19 May 2023
Published: 25 May 2023



## 1. Introduction

Continuously operated liquid/liquid extraction columns are designed to intensively contact two immiscible liquids aiming at the optimization of mass transfer. Agitated extraction columns use mechanical energy to control the droplet size distribution and the dispersed phase holdup. These general design aspects can be transferred to the requirements of multiphase reactions. The simple and effective design of the Taylor-Couette disc contactor (TCDC) is perfectly suited to provide continuous multiphase contact [1]. Modification of the basic contactor design enables contact of a gas phase and liquid phase(s) in co- and countercurrent operation. The phase contact principle of the Taylor-Couette disc contactor is based on stabilized Taylor vortex flow. Gas/liquid contact in Taylor-Couette reactors has been investigated by several authors [2,3]. Stabilization of two-phase flow and the limitation of axial back mixing in the Taylor vortex flow regime using rotor discs or ribs was confirmed by Richter et al. in 2008 [4], resulting in confirmation of the applicability of TCDC columns in continuous countercurrent gas/liquid operations. Continuous gas/liquid contactors such as bubble columns, packed bed or spray columns have been well investigated, and design algorithms are available [5]. These columns make use of gas flow to provide mixing energy. The bubble size distribution and mass transfer area are mainly

controlled by the sparger design. Sufficient mixing without back mixing over the entire volume of the column requires the sophisticated design of a multiphase operation. Large gas/liquid reactors may suffer from insufficient cross-sectional mixing, which significantly limits process efficiency. The TCDC design is supposed to overcome these drawbacks by providing several compartments along the entire column height. Each compartment has the same flow condition depending on the rotational speed of the shaft, enabling uniform dispersed gas phase holdup and mass transfer area along the entire column height. Mixing energy is provided by the rotor shaft and rotor discs. The vertical arrangement of mixing compartments is a multi-stage agitated gas/liquid contactor. These contacting devices are used to provide sufficient residence time combined with minimal axial back mixing even at low gas loads [6]. To demonstrate the applicability of the TCDC in gas/liquid operations, continuous chemisorption of $CO_2$ in 0.1 molar sodium hydroxide solution was performed for different process conditions, aiming at providing a cost-effective $CO_2$ capture technology by chemisorption in sodium hydroxide [7]. The same process can be used to neutralize alkali wastewater streams before further processing. The neutralization of sodium hydroxide in bubble columns is well investigated and design algorithms are available [8–11]. Continuous countercurrent neutralization of 0.1 molar sodium hydroxide solution in the TCDC was performed using a gas mixture of 30 vol% $CO_2$ and 70 vol% $N_2$. The experimental parameters of flow rate, temperature (25 °C, 40 °C, 60 °C) and rotational speed were varied. For modelling of the chemisorption process over column height, the residence time distribution of the continuous liquid phase, the dispersed gas phase holdup, the pH value at the column outlet and the $CO_2$ concentration at the column outlet were monitored. The performance and modelling of a lab-size TCDC column in reactive gas/liquid operations is reported. The continuous chemisorption of $CO_2$ in 0.1 molar sodium hydroxide solution was chosen for investigating gas/liquid phase contact in the TCDC.

## 2. Materials and Methods

### 2.1. Chemisorption of $CO_2$

According to Pohorecki et al. [12], the absorption of $CO_2$ into aqueous alkaline solutions is divided into three steps (Equations (1)–(3)). Reaction 1 represents the physical absorption of gaseous $CO_2$ into the liquid solution. Due to the high rate of absorption equilibrium, concentration at the interface can be assumed. In reaction 2, the dissolved carbon dioxide reacts with hydroxide ions to form the intermediate hydrogen carbonate ($HCO_3^-$). The rate of reaction 3, where the intermediate $HCO_3^-$ reacts with hydroxide ions ($OH^-$) to carbonate ($CO_3^{2-}$), is significantly higher than the rate of reaction 2. Therefore, the rate of the second-order reaction (reaction 2) is defined as a rate-controlling step for chemisorption of $CO_2$ in 0.1 molar sodium hydroxide solution. For a high pH, the pH-depending equilibrium concentration of $HCO_3^-$ is very small (Figure 1). According to Fleischer et al. [13], the reaction of $CO_2$ with water to form bicarbonate and hydrogen ions becomes dominant at low pH and can be neglected in this study because a steady-state pH value at the liquid phase outflow at the column outlet was adjusted to pH 9. At high pH values, absorption of $CO_2$ is considered as rate controlling [8,13,14].

$$CO_{2(g)} \rightleftharpoons CO_{2(l)} \tag{1}$$

$$CO_{2(l)} + OH^- \rightarrow HCO_3^- \tag{2}$$

$$HCO_3^- + OH^- \rightleftharpoons CO_3^{2-} + H_2O \tag{3}$$

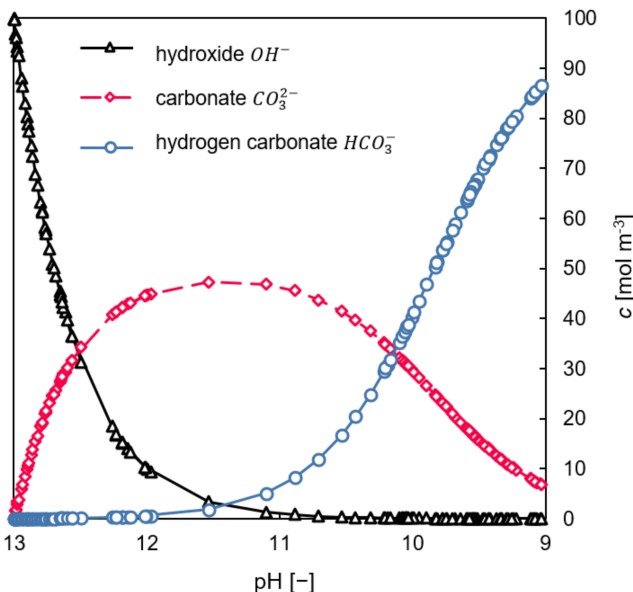

**Figure 1.** Concentration of ion species over decreasing pH value calculated using the Henderson–Hasselbach equation and pKa value of 10.14. The pH value was measured at the column outlet for the following experimental conditions: 25 °C, 300 rpm, 1 L $\text{min}_{\text{STP}}^{-1}$ of 30 vol% $CO_2$ and 70 vol% $N_2$ gas mixture and 0.18 mL $\text{min}^{-1}$ gas/liquid flow and 0.1 molar NaOH in the feed.

For this study, reactions 1 to 3 are considered. Concentration of $CO_3^{2-}$ and $HCO_3^-$ are needed for further calculations of the reaction rate. The ratio between the hydrogen carbonate and the carbonate species is calculated using the Henderson–Hasselbach equation (Equation (4)), as deduced from the law of mass action for step 3. The concentration related pKa value of $CO_2$ in water is affected by ionic strengths and was fitted to close the overall mass balance [15]. The obtained pKa values are in the range between 10.14 and 10.36 depending on the temperature. These results agree well with the literature [14]. Figure 1 illustrates the concentration of ion species over decreasing pH value.

$$pH = pK_a + \log\left(\frac{c_{\text{CO}_3^{2-}}}{c_{\text{HCO}_3^-}}\right) \tag{4}$$

*2.2. Experimental Setup*

A schematic diagram of the experimental setup is given in Figure 2. This lab-scale TCDC DN 50 column is a flexible plant for multiphase operations. The rotor is powered with a laboratory drive M (Heidolph Hei-TORQUE Precision 200). The peristaltic pump P1 (ISMATEC Ecoline VC-280) equipped with Tygon® ELFL 6.4 × 9.6 × 1.6 mm tubing was used as a feed pump for the 0.1 molar NaOH solution which was added on the top of the column. For fixed gas flow rate, the NaOH feed was adjusted via pH control at the column outlet. To monitor continuous neutralization of sodium hydroxide over the column height, the plant was equipped with a pH sensor at the column outlet (SI Analytics BlueLine 22 connected to Knick portavo 904). The flow meter FL (Bio-Tech FCH-m-PP-LC) was used to monitor the NaOH flow rate. A calibrated mass flow controller MFC (Bronkhorst F-201CV) was used to adjust the gas flow rate at the bottom of the column. The $CO_2$ concentration on top of the column was measured using a FlowEvo NDIR sensor (smartGas). For hydraulic characterization, a pressure differential transducer PDT (ICM 331, Schneider Messtechnik, Bad Kreuznach, Germany) was used to monitor the dispersed gas phase holdup. The oxygen concentration was recorded by WTW CellOx 325 sensors at the column outlet ($O_2$1) and in the feed tank ($O_2$2). To ensure constant temperatures between 25 °C and 60 °C, the column was adjusted with a double jacket and a Julabo DYNEO DD-600F (C1) temperature controller. The NaOH feed was preheated with a

glass heat exchanger (HE) connected to the Julabo CORIO CD-601F (C2) controller before entering the column. The double-jacket feed tank B1 was connected to a second Julabo CORIO CD-601F controller. The temperature along the active column was measured using the upper TI1 probe (MGW Lauda R 24/2) and lower TI2 probe (MGW Lauda R 40/2). For neutralization experiments, a gas mixture of 30 vol% $CO_2$ and 70 vol% $N_2$ (Air Liquide) was used. Hydraulic characterization and mass transfer experiments were performed with synthetic air. Nitrogen (Air Liquide, 99.8%) was used for oxygen desorption in mass transfer experiments. Measurement of the residence time distribution was performed using a custom-made conductivity probe that can be installed at the pH port. The NaOH tracer was injected into the NaOH inlet port. The geometric data of the TCDC column are shown in Table 1 and illustrated in Figure 3 (right) according to the design recommendations of Aksamija et al. [16]. Detailed information about the rotor discs is given in Section 2.3.

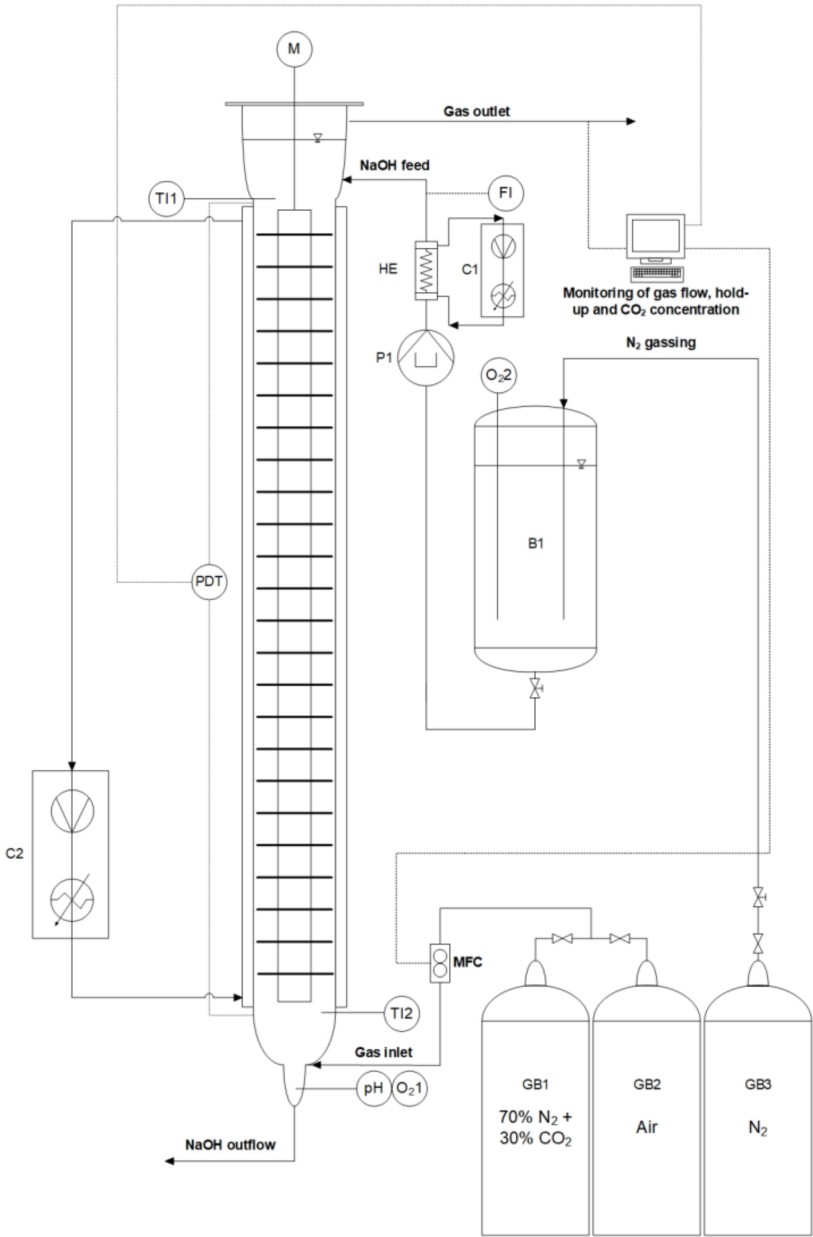

**Figure 2.** PID sketch of the experimental setup. The setup includes the holdup measurement with differential pressure transducer (PDT) and the oxygen probes ($O_2$1 and $O_2$2) for $k_La$ determination. The $CO_2$ sensor on top of the column and the pH probe at the bottom of the column were used to monitor the chemisorption process in neutralization experiments.

**Table 1.** Geometric data of the lab-scale TCDC DN 50 column.

| TCDC DN50 | | | |
|---|---|---|---|
| active length | H | 0.875 | [m] |
| column diameter | $d_C$ | 0.05 | [m] |
| shaft diameter | $d_{Sh}$ | 0.025 | [m] |
| number of compartments | N | 24 | [-] |
| compartment height | $H_C$ | 0.025 | [m] |
| column volume | $V_C$ | 1.35 | [l] |
| rotor disc diameter | $d_R$ | 0.043 | [m] |

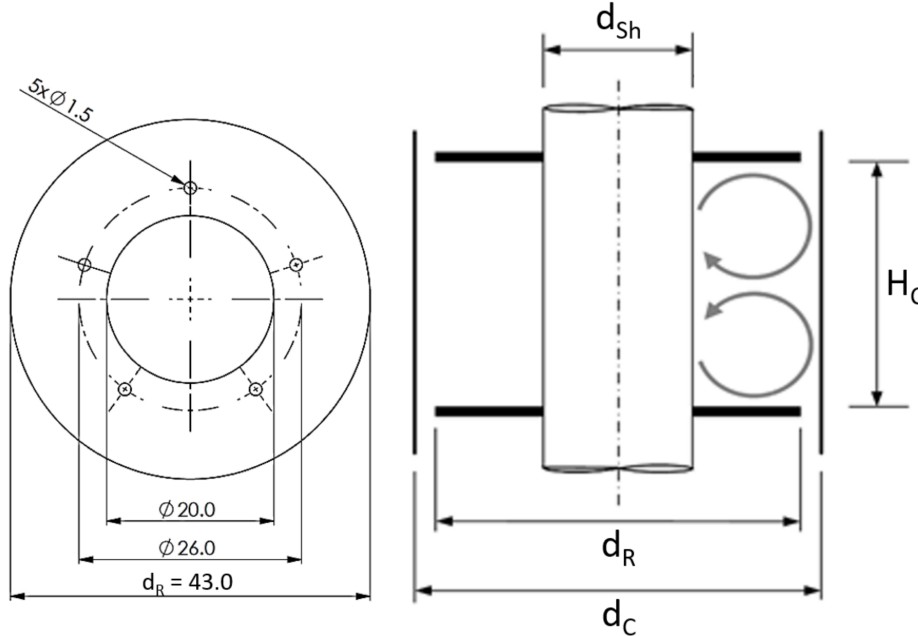

**Figure 3.** Left: Perforated rotor disc for gas/liquid contact. The rotor discs are made of stainless steel with a thickness of 1 mm. All dimensions are given in mm. Right: schematic drawing of a single compartment. Dimensions are summarized in Table 1.

### 2.3. Rotor Disc Design for Gas/Liquid Contact

Controllable dispersed gas phase holdup requires adjustment of the rotor disc design. Therefore, the rotor discs are perforated with 5 holes with a diameter of 1.5 mm, which are located next to the rotor shaft. These holes come with the effect of decreasing dispersed gas phase holdup over increasing rotational speed. The rotor disc design is illustrated in Figure 3 (left).

### 2.4. Column Hydrodynamics
#### 2.4.1. Dispersed Gas Phase Holdup

The dispersed gas phase holdup $\varphi_g$ is defined as the volume fraction of dispersed gas phase bubbles $V_g$ in the active reactor height and the total active reactor volume. It is calculated according to Equation (5).

$$\varphi_g = \frac{V_g}{V_g + V_l} \tag{5}$$

The static pressure equilibrium between a reference tube, filled with the continuous phase, and the aerated column (PDT in Figure 2) corresponds to Equation (6).

$$\rho_l \cdot g \cdot h = \Delta P + g \cdot h \cdot (\varphi_g \cdot \rho_g + (1 - \varphi_g) \cdot \rho_l) \tag{6}$$

The dispersed gas phase holdup can be calculated from the differential pressure $\Delta P$ measured with the differential pressure transducer at steady-state operation. Because of the high-density difference between the gaseous and aqueous phase, the gas phase density can be neglected for holdup calculations. The dispersed gas phase holdup can be simplified to Equation (7).

$$\varphi_g = \frac{\Delta P}{g \cdot h \cdot \rho_l} \cdot 100 \tag{7}$$

### 2.4.2. Residence Time Distribution

Residence time distribution is analyzed by measuring the response signal of the conductivity sensor at the column outlet after tracer injection on top of the column. For this study, spiking of the aqueous phase with 0.5 mL of saturated sodium chloride was used for residence time distribution measurements. Calculation of the exit age function $E_i$ according to the measured relative conductivity $g_i$ is performed according to Equation (8). The measured mean residence time is given by Equation (9).

$$E_i = \frac{g_i}{\sum(g_i \cdot \Delta t_i)} \tag{8}$$

$$\bar{t} = \frac{\sum(t_i \cdot g_i \cdot \Delta t_i)}{\sum(g_i \cdot \Delta t_i)} \tag{9}$$

The exit age function and the time are normalized to the dimensionless exit age function $E_{\theta i}$ and the dimensionless time $\theta$ with Equations (10) and (11).

$$\theta_i = \frac{t_i}{\bar{t}} \tag{10}$$

$$E_{\theta_i} = \frac{E_i}{\bar{t}} \tag{11}$$

According to Octave Levenpiel [17], large deviation from ideal plug-flow behavior can be approximated using the CSTR cascade model. The number of CSTRs in series is obtained from $E_{\theta,max}$ according to Equation (12).

$$E_{\theta,max} = \frac{N \cdot (N-1)^{N-1}}{(N-1)!} \exp{-(N-1)} \tag{12}$$

The dispersion model uses the Bodenstein number (Bo) as an indicator for axial back mixing. For large deviation from ideal flow and open-open vessel boundary condition, Bo can be calculated from the variance $\sigma$ of the E function according to Equation (13) [17].

$$Bo = \frac{1}{\sigma_{E\theta_i}^2} + \sqrt{\left(\frac{1}{\sigma_{E\theta_i}^2}\right)^2 + \frac{8}{\sigma_{E\theta_i}^2}} \text{ with } \sigma^2 = \int_0^\infty (t - \bar{t})^2 \cdot E(t) \cdot dt \tag{13}$$

The ratio between convective flow and axial dispersion ($D_{ax}$) is defined by the Bodenstein number according to Equation (14).

$$Bo = \frac{u_l \cdot L}{D_{ax}} \tag{14}$$

### 2.4.3. Liquid Phase Volumetric Mass Transfer Coefficient $k_L a$

The experimental procedure to determine the volumetric mass transfer coefficient of oxygen in water is described by Skala and Veljkovic [18]. The dissolved oxygen concentration is measured at the column inlet ($O_2 2$ in Figure 2) and outlet ($O_2 1$). The data of both sensors were recorded using a custom LabVIEW routine. A continuous steady-state measurement of oxygen concentration for specific experimental conditions (temperature,

flow rate, rotational speed) requires successive absorption and desorption in a loop. Therefore, the feed tank was bubbled with nitrogen. The nitrogen flow rate in the feed tank was adjusted to keep the dissolved oxygen concentration at the column inlet below 1 mg L$^{-1}$. Synthetic air was then added to the column and the dissolved oxygen concentration in water was measured at the column outlet at steady state. The $k_La$ value was calculated according to Equation (15). The determination at steady-state mass transfer conditions in countercurrent flow regimes excludes errors resulting from low-sensitivity oxygen probes. This method is also described by Chaumat et al. [19] and is successfully used for investigating mass transfer in bubble columns [20].

$$k_{La} = \frac{u_l}{H} \cdot \ln \frac{c^* - c_{out}}{c^* - c_{in}} \tag{15}$$

The superficial phase velocity u of a single phase is related to the free cross-sectional area between the rotor shaft and the column wall $A_{ring}$ (Equation (16))

$$u = \frac{\dot{V}}{A_{ring}} \tag{16}$$

For specified temperature and pressure, the equilibrium concentration c* at the middle of the column was calculated using Equation (17), according to Schumpe et al. [21]. The temperature was measured simultaneously using WTW CellOx325 sensors. The pressure P in the middle of the column was calculated from the column height and measured dispersed phase holdup. The column outlet is open to the atmosphere and the pressure at the column outlet was set to ambient pressure.

$$c^* \left[ \frac{mg\ O_2}{L} \right] = 2.954 \cdot (P - P_s) \cdot \alpha \tag{17}$$

The pressure P in Equation (20) is given in kPa and $\alpha$ was calculated according to Equation (18) with the temperature T in °C. Coefficients a to e are given in Table 2 [21]. The coefficients are defined for oxygen solubility in a temperature range from 0 °C to 50 °C.

$$\alpha = a + b \cdot T + c \cdot T^2 + d \cdot T^3 + e \cdot T^4 \tag{18}$$

**Table 2.** Coefficients of the power series to calculate $O_2$ solubility with Equation (18) as a function of temperature (0–50 °C) [21].

| Parameter | Value |
|:---:|:---:|
| a | $4.900 \cdot 10^{-2}$ |
| b | $-1.335 \cdot 10^{-3}$ |
| c | $2.759 \cdot 10^{-5}$ |
| d | $3.235 \cdot 10^{-7}$ |
| e | $1.614 \cdot 10^{-9}$ |

The saturation pressure $P_s$ in Equation (17) was calculated according to the Antoine equation (Equation (19)) with T given in K and the pressure P in bar. Antoine parameters for the temperature range from 293 K to 343 K are: A = 6.20963, B = 2354.731 and C = 7.559 [22].

$$\log_{10}(P_s) = A - \frac{B}{T + C} \tag{19}$$

### 2.5. Reactor Modelling

Heterogeneous chemisorption of $CO_2$ in aqueous NaOH solution has been studied by many authors. The design algorithm proposed by Darmana et al. [11] is commonly

used to describe the chemisorption of $CO_2$ in bubble columns [8,10,14,23,24]. Therefore, the neutralization of NaOH with $CO_2$ is modelled in respect to $CO_2$ consumption according to the two-film theory. Equation (20) shows the general reaction equation for component A ($CO_2$) being transferred from a gaseous phase to a liquid bulk phase and consecutive reaction with component B ($OH^-$) to form product C ($CO_3^{2-}$) whereas the stoichiometric factor b = 2. This reaction is a second-order irreversible reaction as reactants B and C are not soluble in the gaseous phase [24]. Octave Levenspiel defined different reaction regimes in respect to the rate-limiting step. The general reaction rate for gas/liquid reactions according to Levenspiel is expressed in Equation (21) [17].

$$A_{(g \to l)} + bB_{(l)} \to C_{(l)} \tag{20}$$

$$-r_A'''' = \frac{p_A}{\frac{1}{k_{AG}a} + \frac{1}{He_A \cdot k_{AL}a \cdot E} + \frac{1}{He_A \cdot k_A \cdot c_B \cdot (1 - \varphi_g)}} \tag{21}$$

The gas solubility of $CO_2$ in the liquid phase, dependent on the partial pressure p, is described by Henry's law (Equation (22)).

$$c_{CO_2,l} = p_{CO_2} \cdot H_{CO_2} \tag{22}$$

The Henry constant of $CO_2$ in pure water $H_{w,CO_2}$ [mol m$^{-3}$ Pa$^{-1}$] is calculated according to Equation (23) with the temperature T in K [25].

$$H_{w,CO_2} = 3.54 \cdot 10^{-7} \exp\left(\frac{2044}{T}\right) \tag{23}$$

The effect of electrolyte solutions on the Henry constant is taken into account by Equation (24), with ci indicating the ion concentration [26].

$$\log\left(\frac{H_{w,CO_2}}{H_{CO_2}}\right) = \sum (hi + hg) \cdot ci \tag{24}$$

Ion specific parameters for hi and hg are summarized in Table 3.

**Table 3.** Ion specific parameters according to Weisenberger and Schumpe [26].

| Ion | hi [m³ kmol⁻¹] | Gas | hg [m³ kmol⁻¹] | bi [m³ kmol⁻¹] |
|---|---|---|---|---|
| $Na^+$ | 0.1143 | $CO_2$ | −0.0172 | −0.0857 |
| $OH^-$ | 0.0839 | | | −0.1088 |
| $HCO_3^-$ | 0.0967 | | | −0.115 |
| $CO_3^{2-}$ | 0.1423 | | | −0.245 |

According to Pohorecki and Moniuk [12], the reaction rate of the second-order reaction rate constant $k_{OH^-}$ [m³ kmol⁻¹ s⁻¹] for the chemisorption of $CO_2$ into aqueous NaOH solution is influenced by temperature and ionic strength. Therefore, the reaction rate constant at infinite solution $k_\infty$ is calculated according to Equation (25) and the ionic strength is considered with Equations (26) and (27) [12].

$$\log k_\infty = 11.895 - \frac{2382}{T} \tag{25}$$

$$\log\left(\frac{k_{OH^-}}{k_\infty}\right) = 0.221 \cdot I - 0.016 \cdot I^2 \tag{26}$$

Whereas, the ionic strength I is defined as:

$$I = 0.5 \cdot (c_{Na^+} \cdot z_{Na^+}^2 + c_{OH^-} \cdot z_{OH^-}^2 + c_{HCO_3^-} \cdot z_{HCO_3^-}^2 + c_{CO_3^{2-}} \cdot z_{CO_3^{2-}}^2) \tag{27}$$

Gas/liquid reactions are mainly affected by the diffusion of the reactants. Mass transfer of reactants is enhanced when the transferred component is consumed by the chemical reaction. This increased mass transfer can be described by the enhancement factor E, which describes the ratio of mass transfer with and without chemical reaction. The Hatta number Ha (Equation (28)) and the enhancement factor for an instantaneous reaction $E_\infty$ (Equation (29)) are used to categorize the reaction regime [17,24,27]. For this study, the approximation according to Westerterp et al. [28] is used for calculation of the enhancement factor E (Equation (30)) [8,11,24,29].

$$\text{Ha} = \frac{\sqrt{k_{OH^-} \cdot D_{CO_2,l} \cdot c_{OH^-,l}}}{k_{L,CO_2}} \tag{28}$$

$$E_\infty = 1 + \left( \frac{D_{OH^-,l} \cdot c_{OH^-,l}}{2 \cdot D_{CO_2,l} \cdot c_{CO_2,l}} \right) \cdot \left( \frac{D_{CO_2,l}}{D_{OH^-,l}} \right)^{\frac{1}{2}} \tag{29}$$

$$E = \begin{cases} 1 & E_\infty \leq 1 \\ -\frac{\text{Ha}^2}{2(E_\infty - 1)} + \left( \frac{\text{Ha}^4}{4(E_\infty - 1)^2} + \frac{E_\infty \cdot \text{Ha}^2}{E_\infty - 1} + 1 \right)^{\frac{1}{2}} & 1 < E_\infty \end{cases} \tag{30}$$

For the calculation of Ha and E, further parameters are needed. Therefore, the diffusion coefficient of pure $CO_2$ in water $D_{w,CO_2} \; [\text{m}^2 \, \text{s}^{-1}]$ was calculated using Equation (31) [25].

$$D_{w,CO_2} = 2.35 \cdot 10^{-6} \exp\left( \frac{-2119}{T} \right) \tag{31}$$

The influence of electrolyte solutions on the diffusion coefficient is considered with Equation (32). Ion specific parameters for bi are listed in Table 3 [30].

$$D_{CO_2} = D_{w,CO_2} \cdot \left( 1 + 0.624 \cdot \sum b_i \cdot c_i \right) \tag{32}$$

Temperature-dependent diffusion coefficients for $OH^-$ ions $D_{OH^-} \; [\text{m}^2 \, \text{s}^{-1}]$ are calculated according to Equation (33) [31]. The parameters are summarized in Table 4.

$$D_{OH^-} = D_{OH^-}^0 \cdot \left( \frac{T}{T_0} - 1 \right)^{\psi} \tag{33}$$

**Table 4.** Parameters for determination of $OH^-$ diffusivity [24].

| $\psi$ | $D_{OH^-}^0 \; [\text{m}^2 \, \text{s}^{-1}]$ | $T_0 \; [K]$ |
|---|---|---|
| 1.658 | $2.665 \cdot 10^{-8}$ | 216.5 |

### 2.6. Chemicals

The chemicals used for the neutralization experiments were technical grade according to Table 5.

**Table 5.** Chemicals used for neutralization experiments and hydraulic characterization.

| Name | Manufacturer | CAS | Purity |
|---|---|---|---|
| sodium hydroxide | J.T. Baker | 1310-73-2 | ≥98% |
| deionized water | in-house source | | |
| synthetic air | Air liquide | | ≥99% |
| nitrogen | Air liquide | 7727-37-9 | ≥99.8% |
| gas mix 30% $CO_2$ and 70% $N_2$ | Air liquide | | ≥99% |

## 3. Results

### 3.1. Neutralization Experiments

The 0.1 molar NaOH feed was prepared in a 10 L container by dissolving 40 g of solid NaOH pellets in deionized water. The measured start pH value of 0.1 molar NaOH solution at ambient temperature was pH 13. Calibration of the pH probe was performed between pH 7 and pH 12 prior to each neutralization experiment. Knick CaliMat buffer solutions were used for calibration. The steady-state pH value at the column outlet was set to pH 9 to avoid further reaction of $CO_2$ with water. The gas flow rate was fixed to 1, 2 and 3 L $\text{min}_{STP}^{-1}$ (30 vol% $CO_2$ and 70 vol% $N_2$). The 0.1 molar NaOH flow rate was experimentally adjusted to obtain a constant pH of 9 at the column outlet. The entire set of neutralization experiments for different temperatures and rotational speeds is listed in Table 6.

**Table 6.** List of neutralization experiments.

| T | rpm | $\dot{V}_g$ | $\dot{V}_l$ |
|---|---|---|---|
| [°C] | [L min$^{-1}$] | [L min$_{STP}$$^{-1}$] | [L min$^{-1}$] |
| 25 | 300 400 700 | 1 | 0.18 |
| 25 | 300 400 700 | 2 | 0.28 |
| 25 | 300 400 700 | 3 | 0.42 |
| 40 | 300 400 700 | 1 | 0.18 |
| 40 | 300 400 700 | 2 | 0.28 |
| 40 | 300 400 700 | 3 | 0.42 |
| 60 | 300 400 700 | 1 | 0.18 |
| 60 | 300 400 700 | 2 | 0.28 |
| 60 | 300 400 700 | 3 | 0.42 |

The experimentally obtained molar ratio between the NaOH feed and $CO_2$ is shown in Figure 4 and is approached with a second-order polynomial (Equation (34)). The NaOH feed was adjusted to 25 °C at a rotational speed of 300 rpm and was kept constant for all temperatures and stirrer speeds.

$$\dot{n}_{CO_2} = -0.26 \cdot \dot{n}_{NaOH}^2 + 2.07 \cdot \dot{n}_{NaOH} - 1.12 \tag{34}$$

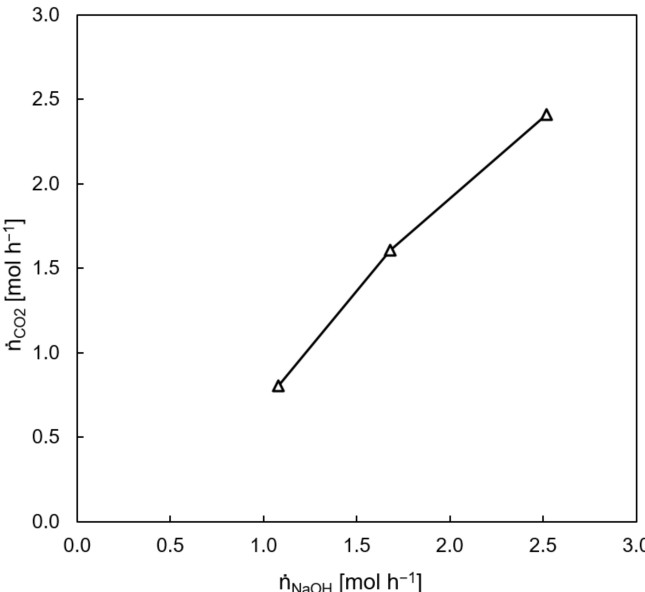

**Figure 4.** Molar ratio of $CO_2$ vs. NaOH feed for constant pH of pH 9 at the column outlet.

During column startup, the column outlet was connected to the feed tank and the NaOH feed was set to 60 mL min$^{-1}$ until constant flow and constant temperatures were reached. Column startup at a low NaOH feed rate was performed in order to decrease the spent amount of sodium hydroxide and to provide comparability of startup at different process parameters. The stirrer speed was adjusted to the desired rotational speed and the gas phase was set to the desired flow rate. After adding $CO_2$, the neutralization reaction started, and the aqueous phase column outlet was connected to the waste tank. During neutralization experiments, the feed tank was permanently refilled with 0.1 molar NaOH. The pH value at the column outlet decreased over time. After reaching pH 9, the NaOH feed was adjusted to the experimental conditions given in Table 6. Figure 5 illustrates this process with the overswing of the pH and $CO_2$ outlet concentration after increasing the NaOH feed. At steady-state operation, the entire amount of $CO_2$ was consumed by the neutralization reaction.

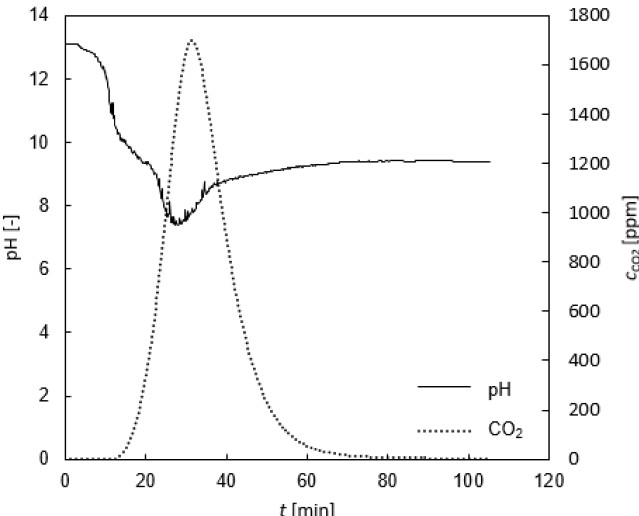

**Figure 5.** Measured pH value at the column outlet over time and corresponding $CO_2$ concentration in ppm in the exhaust gas for a reaction temperature of 25 °C and gas feed of 1 L min$_{STP}$$^{-1}$ (30 vol% $CO_2$ and 70 vol% $N_2$); stirrer speed 300 rpm and 60 mL min$^{-1}$ 0.1 molar NaOH feed rate until pH 9 was obtained. At pH 9, the NaOH feed was changed to 180 mL min$^{-1}$.

### 3.1.1. Influence of Rotational Speed

Figure 6 shows the pH over time during the column startup. The rotational speed was varied at a constant temperature of 25 °C and constant gas flow rate. It was expected that the changing flow regime with decreasing dispersed phase holdup (Figure 12), and thus also the decreasing mass transfer area, would show a different trend of decreasing pH, which was actually not monitored.

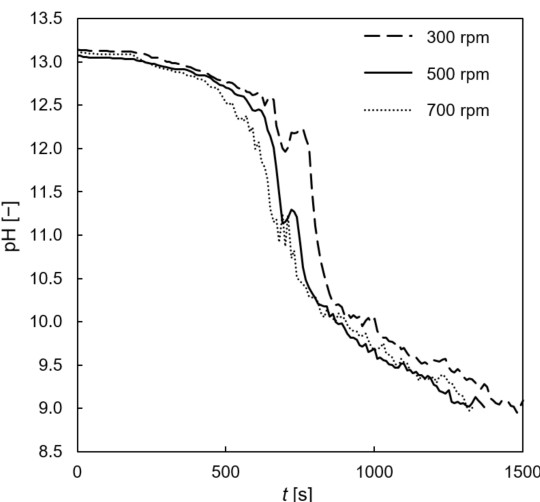

**Figure 6.** Measured pH value at column outlet over time for different rotational speeds with a 60 mL min$^{-1}$ 0.1 molar NaOH feed at 25 °C and 1 L min$_{STP}$$^{-1}$ of gas mixture (30 vol% $CO_2$ and 70 vol% $N_2$).

### 3.1.2. Influence of the Reaction Temperature

Figure 7 shows the influence of temperature on pH during column startup. The decreasing starting pH is explained by the strongly temperature-dependent dissociation of water [32]. The temperature-dependent pH value makes it more difficult to directly compare the neutralization process. Steady-state operation at the desired pH of 9 was reached earlier at elevated temperatures, which agrees with the temperature-dependent reaction rate constant in Equation (2) [12]. The concentration of sodium hydroxide was 0.1 molar for all experiments.

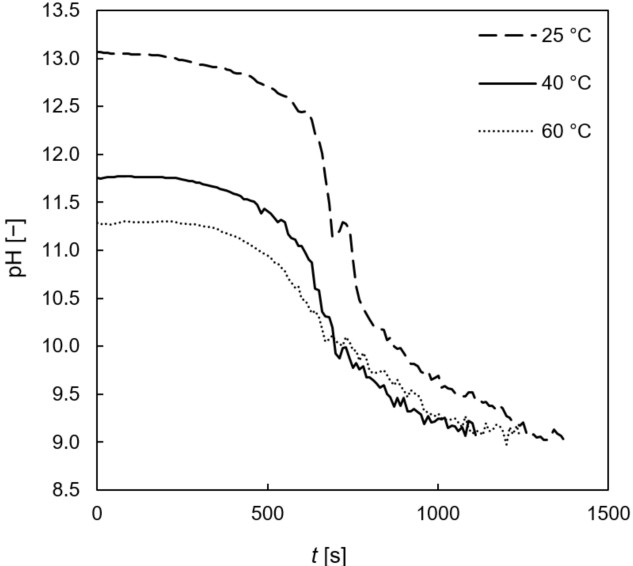

**Figure 7.** Measured pH at the column outlet over time for different temperatures with a 60 mL min$^{-1}$ feed rate of 0.1 molar NaOH, 500 rpm and 1 L min$_{STP}$$^{-1}$ (30 vol% $CO_2$ and 70 vol% $N_2$) gas flow rate.

### 3.1.3. Influence of Gas Flow Rate

Figure 8 shows the influence of gas flow rate on the pH during column startup. The general expected trend shows a faster pH decrease at higher gas flow rate. With increasing gas flow rate, the dispersed gas phase holdup also increases and therefore, the specific mass transfer area significantly increases.

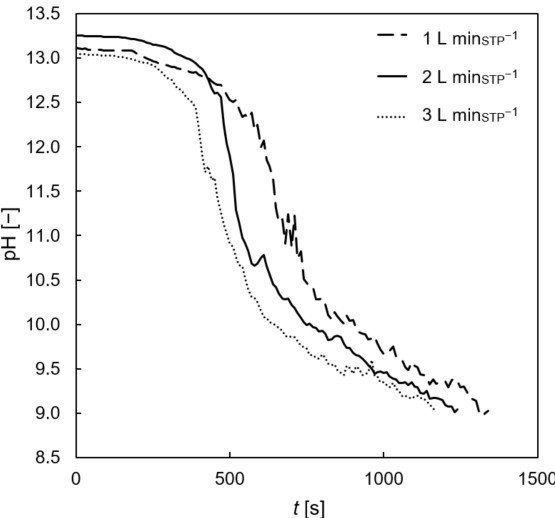

**Figure 8.** Measured pH at the column outlet over time for different gas flow rates (30 vol% $CO_2$ and 70 vol% $N_2$) and 60 mL min$^{-1}$ 0.1 molar NaOH feed, 700 rpm and 25 °C.

### 3.1.4. Steady-State Operation

For the neutralization experiments, steady-state pH is defined for the operation state after flow rate adjustment and constant pH value at the column outlet for at least 15 min. Although the influence of the rotational speed is identified as not significant, the steady-state pH was averaged in terms of rotational speed. Figure 9 shows the dependence of steady-state pH on gas flow rate and temperature. The lowest pH values were reached at a gas flow rate of 2 L min$_{STP}$$^{-1}$. The corresponding NaOH feed rate for each gas flow rate is shown in Table 6.

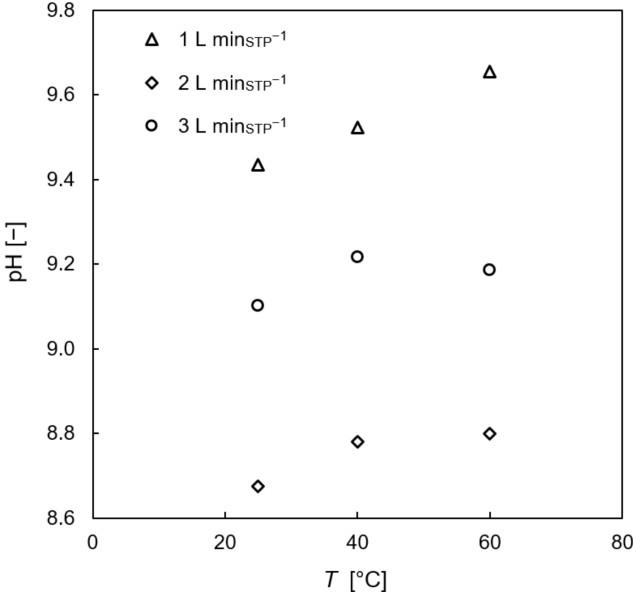

**Figure 9.** Influence of temperature on rpm-averaged pH at the column outlet at steady-state operation.

As shown in Figure 5, the $CO_2$ concentration at the column outlet was continuously recorded. Rpm-averaged steady-state $CO_2$ outlet concentrations are shown in Figure 10. The starting concentration of 30 vol% $CO_2$ was almost depleted to 0 ppm for 1 L $min_{STP}^{-1}$. A maximum $CO_2$ outlet concentration of 490 ppm was reached at 25 °C and 3 L $min_{STP}^{-1}$. The reactor enabled at least 99.9% efficiency with respect to $CO_2$ consumption.

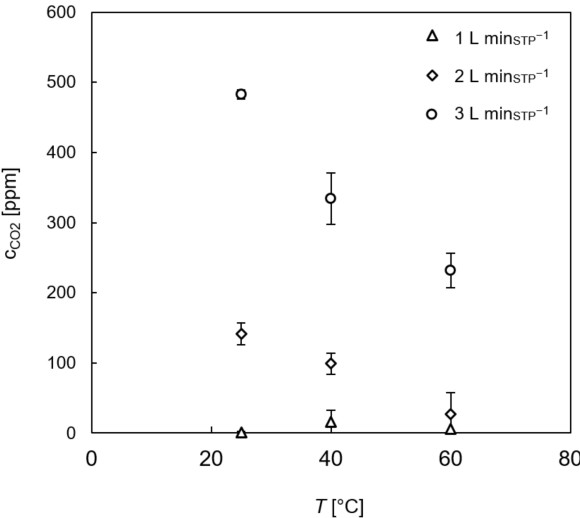

**Figure 10.** Rpm-averaged $CO_2$ concentration at the column outlet for different temperatures and different dispersed gas flow rates. Error bars show the influence of rotational speed.

### 3.2. Column Hydrodynamics

Characterization of column hydrodynamics was performed with the non-reactive test system using deionized water and synthetic air. The process parameters were the same as those used for the neutralization experiments (Table 6).

### 3.2.1. Dispersed Gas Phase Holdup

The perforated rotor discs (Figure 3: left) enable uniform dispersed gas phase holdup over the entire reactor height. At low rotational speed (0 rpm to 300 rpm), the gas phase rises in the gap between the column wall and the rotor disc which is specified as external flow (Figure 11a). Between 300 rpm and 700 rpm, the gas phase partially passes through the 1.5 mm holes and the gap between the rotor disc and the column wall. This flow regime is characterized as mixed flow (Figure 11b). Beyond 700 rpm, the flow regime is defined as internal only because the entire gas phase passes up through the perforated rotor discs (Figure 11c). This characterization of gas/liquid flow in the TCDC is valid for gas flow rates up to 3 L $min_{STP}^{-1}$ and a moderate liquid flow rate of 0.18 L $min^{-1}$ up to 0.42 L $min^{-1}$ as applied in this study. The general trend of dispersed gas phase holdup $\varphi_g$ in the TCDC column is shown in Figure 12. The area of linearly decreasing dispersed phase holdup is used during neutralization experiments. Increasing rotational speed increases the centrifugal forces on the gaseous and liquid phase. The light gaseous phase is pressed towards the rotor shaft, forcing the gas phase to pass up through the perforated rotor discs. This unique flow behavior is used to establish constant dispersed gas phase holdup in the TCDC. Videos of Figure 11 (a) S1, (b) S2 and (c) S3 can be downloaded from Supplementary Material.

The dispersed gas phase holdup was measured using a differential pressure transducer. For all experiments, an IDM 331 (Schneider Messtechnik, 0 mbar to 20 mbar, $\leq\pm1\%$ FSO) was used. According to the static pressure equilibrium (Equation (7)), $\Delta P$ is the measured pressure difference between the bypass line filled with continuous phase (water) and the aerated column. Figure 2 illustrates the connection ports of the differential pressure transducer (PDT). In order to indicate the dynamic pressure caused by the rotation of the rotor shaft, reference measurements were performed without gas flow at the respective

rotational speed and water flow rate. The measured pressure difference without aeration was subtracted from the differential pressure measured at aerated conditions. Holdup measurements were performed at 25 °C. The results are shown in Figure 13. The dispersed gas phase holdup $\varphi_g$ in the TCDC setup under experimental conditions was between 2% and 16% depending on the rotational speed and gas flow rate.

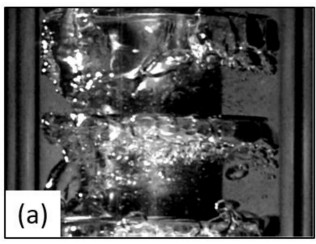 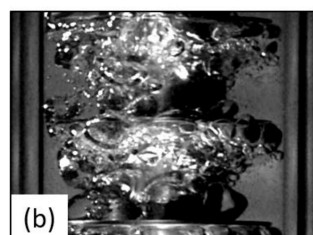 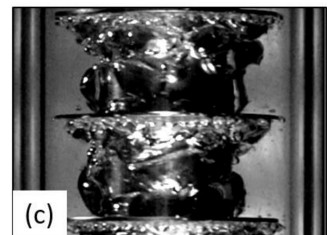

**Figure 11.** Changing flow regimes for different rotational speeds at a constant gas feed rate of 2 L min$_{STP}$$^{-1}$ and liquid feed flow rate of 0.28 L min$^{-1}$; (**a**) at 300 rpm (mainly external flow); (**b**) mixed flow at 500 rpm; (**c**) internal flow only at 700 rpm. Videos of flow regime (**a**) S1, (**b**) S2 and (**c**) S3 are included in the Supplementary Materials.

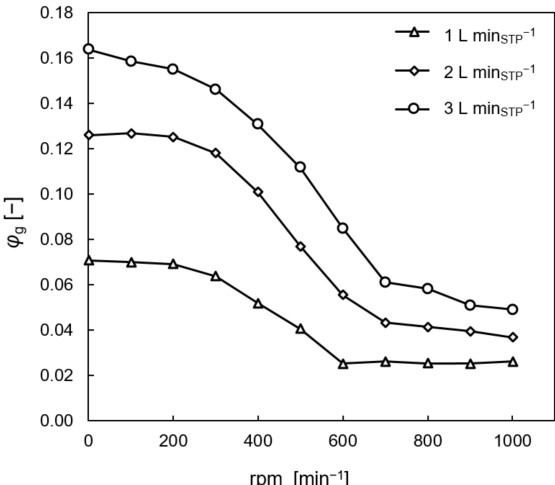

**Figure 12.** Dispersed gas phase holdup $\varphi_g$ in the TCDC for a constant water flow rate of 245 mL min$^{-1}$ and a different flow rate of synthetic air (1 L min$_{STP}$$^{-1}$, 2 L min$_{STP}$$^{-1}$, 3 L min$_{STP}$$^{-1}$).

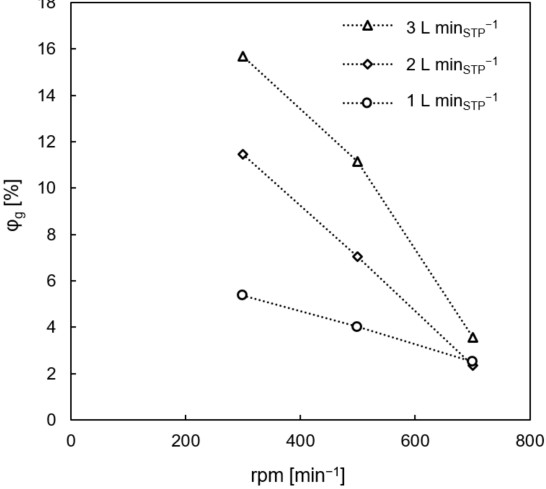

**Figure 13.** Dispersed gas phase holdup and rotational speed at different volumetric flow rates at 25 °C and aqueous phase feed according to Table 6.

3.2.2. Residence Time Distribution

Concentration gradients are affected by unavoidable axial back mixing. For modelling, the CSTR cascade model [1,19] was applied. Pulse experiments were performed at 25 °C for all operating points given in Table 6. The recorded data in normalized form, as described in Section 2.4, were (for convenience) fitted with a four-parameter log-normal distribution (Equation (35)). a, b, c and d are the fit parameters and θ is the dimensionless residence time.

$$
E_{\theta_{i,model}} = a + b \cdot \exp\left(-\frac{1}{2} \cdot \left(\frac{\ln\left(\frac{\theta}{c}\right)}{d}\right)^2\right) \tag{35}
$$

Figure 14 shows the applied evaluation algorithm for the number of continuously stirred tanks in the cascade. The number of CSTRs (Equation (12)) is compared to the outcome of the log normal fit. The CSTR cascade model describes the measured data well. The mean residence time t (Equation (9)), the number of CSTRs in the cascade, the Bodenstein number Bo (Equation (13)) and the axial dispersion coefficient $D_{ax}$ (Equation (14)) are compared in the following section.

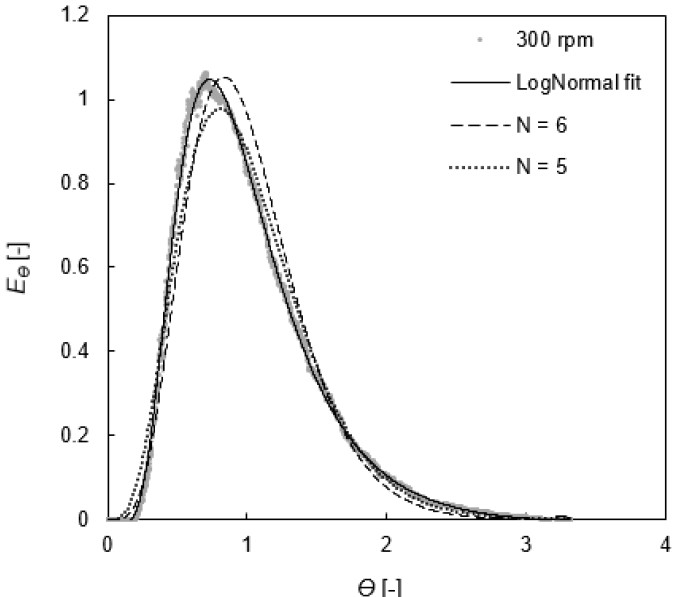

**Figure 14.** Normalized residence time distribution at 300 rpm, 1 L min$_{STP}^{-1}$ air, and 180 mL min$^{-1}$ water. The dots indicate the measured, normalized experimental data. The measured data were fitted with log-normal distribution (solid black line). The dotted line indicates the residence time distribution with 5 CSTRs in series, and the dashed line shows the modelled residence time distribution for 6 CSTRs in series.

Figure 15a shows the measured mean residence times $t_{exp} = \bar{t}$ for the three flow rate parameter sets (lines). The hydraulic residence time $t_{hydro}$ was calculated using Equation (36). The average increase in the residence time compared to the hydraulic residence time was 43%. The influence of rotational speed is small compared to the influence of the flow rate.

$$
t_{hydro} = \frac{V_C}{\dot{V}_l} \tag{36}
$$

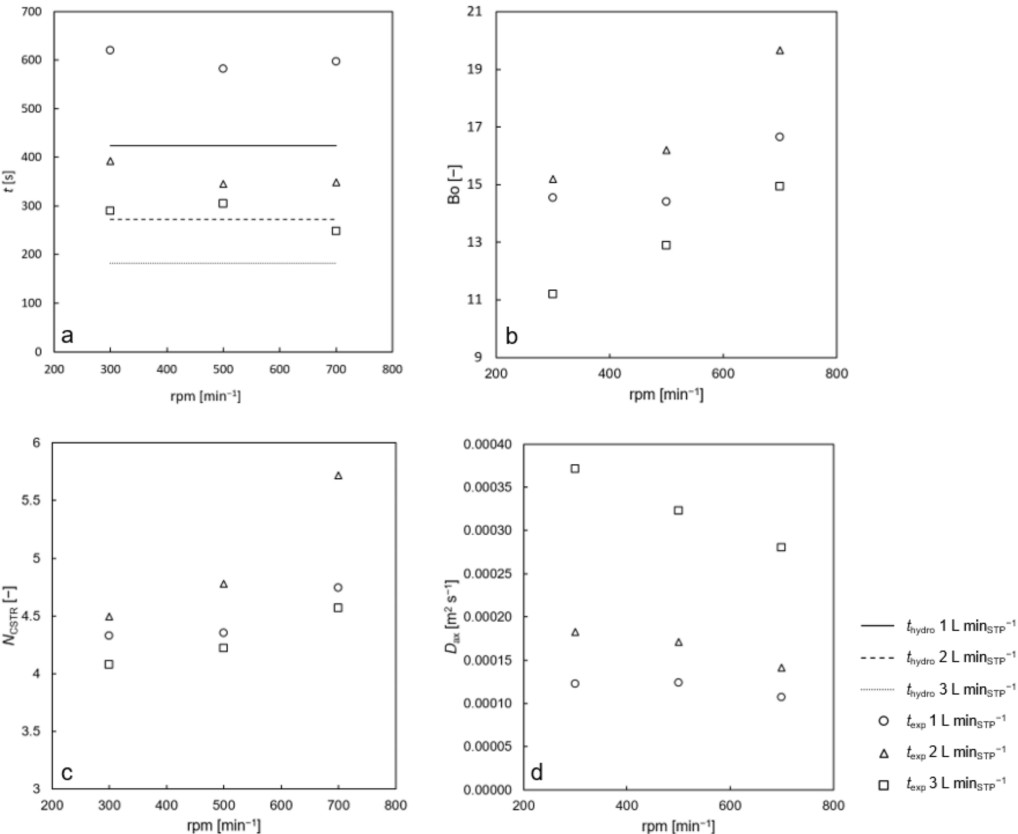

**Figure 15.** Influence of rotational speed at 25 °C on (**a**) mean residence time $t_{exp}$ compared with hydraulic residence time $t_{hydro}$; (**b**) Bodenstein number; (**c**) number of continuously stirred tanks in the cascade $N_{CSTR}$; (**d**) axial dispersion coefficient $D_{ax}$. The corresponding aqueous phase flow rates for the varied gas flow rates are shown in Table 6.

Increasing rotational speed decreases axial back mixing, as shown in Figure 15b, with Bo increasing with increasing rotational speed, increasing $N_{CSTR}$ (Figure 15c) and decreasing axial dispersion coefficient $D_{ax}$ (Figure 15d). Increasing rotational speed in countercurrent gas/liquid contact changes the flow regime, as shown in Figure 4. When the gas flow rate through the perforated rotor discs exceeds the maximum (at about 2 L $\min_{STP}{}^{-1}$), the excess amount of gas is forced to rise in the gap between the rotor discs and the column wall (for example at 3 L $\min_{STP}{}^{-1}$) with a negative effect on back mixing, as shown with the Bo number in Figure 15b, $N_{CSTR}$ in Figure 15c and $D_{ax}$ in Figure 15d.

### 3.2.3. Liquid Phase Volumetric Mass Transfer Coefficient $k_L a$

The liquid side volume-based mass transfer coefficient is determined for steady-state operation conditions with the air (oxygen from air) and water test system as described in Section 2.4. Figure 16 shows little influence of rotational speed on the $k_L a$ value. The influence of rotational speed is small and shows no characteristic trend. However, it confirms the outcome of the RTD measurements. At low gas load (1 L $\min_{STP}{}^{-1}$), $k_{La}$ slightly increases with rotational speed. At 2 L $\min_{STP}{}^{-1}$, we registered a change in the flow regime. In operation state 3 L $\min_{STP}{}^{-1}$, the effect of rotational speed on "bypass gas flow" no longer affected the mass transfer area. Similar results were found by Ramezani et al. who investigated the oxygen mass transfer in an air-water Taylor-Couette reactor [2]. The $k_L a$ value is highly sensitive to the air flow rate (Figure 16) and temperature (Figure 17). $k_L a$ values are in the range of between 0.005 s$^{-1}$ for 25 °C and 1 L $\min_{STP}{}^{-1}$ air flow rate and 0.02 s$^{-1}$ for 60 °C and 3 L $\min_{STP}{}^{-1}$ air flow rate, which agrees well with the literature [2,3].

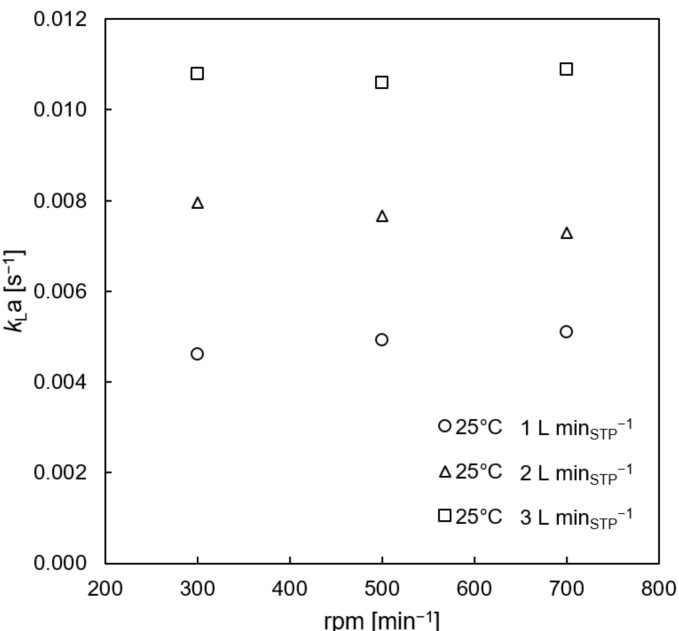

**Figure 16.** Influence of rotational speed on $k_La$ at 25 °C under variation of gas and corresponding liquid flow rates of 0.18 mL min$^{-1}$ for 1 L min$_{STP}$$^{-1}$; 0.32 mL min$^{-1}$ for 2 L min$_{STP}$$^{-1}$ and 0.42 L min$^{-1}$ for 3 L min$_{STP}$$^{-1}$.

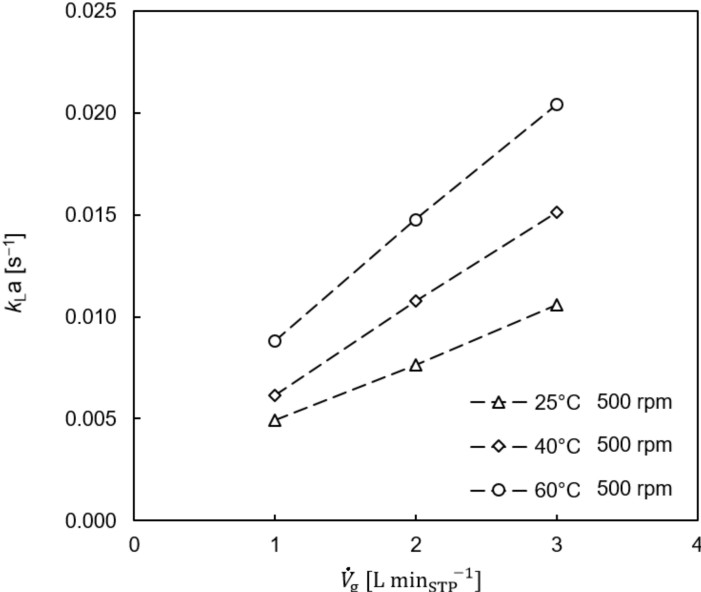

**Figure 17.** Influence of air flow rate $V_g$ on $k_La$ at a rotational speed of 500 rpm. Dotted lines indicate the linear trend of increasing $k_La$ over gas flow rate.

Temperature dependency of the $k_La$ can be described by Equation (37) with the temperature T in °C, measured $k_La$ at 25 °C and the temperature correction factor β fitted for the experimental data [33]. Figure 18 shows the increasing $k_La$ with increasing temperature for specific operation conditions and β = 0.019.

$$k_La = k_La_{25} \cdot \beta^{(T-25)} \tag{37}$$

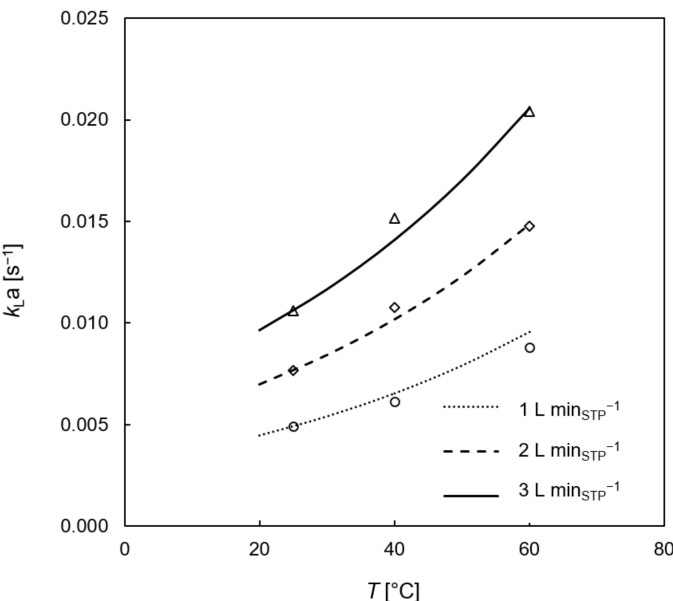

**Figure 18.** Temperature-dependent $k_La$ at 500 rpm. The lines show the fitted temperature dependency with β = 0.019 (Equation (25)). The single markers indicate the experimental data points.

### 3.3. Reactor Modelling

The steady-state experimental data from the neutralization experiments, as discussed in Section 3.1, were used to develop the reactor mass balance. Depending on the consumed $CO_2$ amount in the reactor, the ratio between hydrogen carbonate $HCO_3^-$ and carbonate $CO_3^{2-}$ ions can be calculated by fitting the pKa value of the Henderson–Hasselbach equation (Equation (4)). The measured outlet concentrations of $CO_2$ in the gas phase and $OH^-$ in the aqueous phase are used for the mass balance. Due to the temperature dependency, the pKa value was fitted for the three different reaction temperatures. The literature reports a pKa value of 10.22 at 38 °C [15], which compares well to the fitted value of 10.29 for the neutralization experiments at 40 °C. For 25 °C, a pKa of 10.14 was obtained and for 60 °C, a pKa of 10.36 was obtained. Figure 19 shows a parity plot of the measured and calculated $CO_2$ consumption in the reactor. The fitted pKa values also compare well with thermodynamic pKa values, confirming the quality of the experiments.

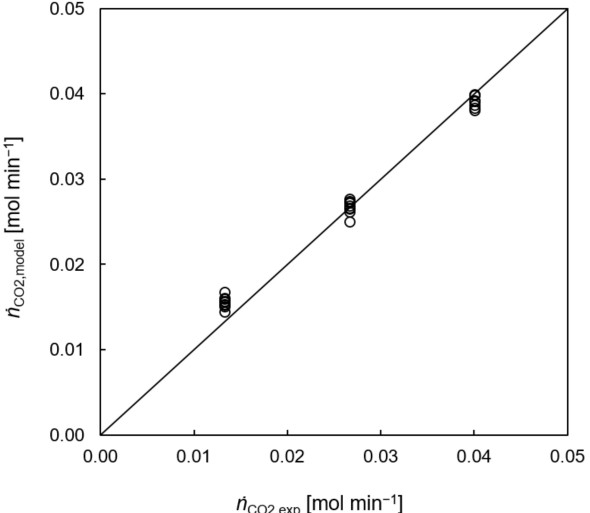

**Figure 19.** Parity plot of the measured $CO_2$ consumption in the reactor vs. the modelled $CO_2$ consumption using the fitted pKa values: 25 °C, pKa = 10.14; 40 °C, pKa = 10.29; 60 °C, pKa = 10.36.

The evaluation of residence time distribution in Section 3.2.2 resulted in at least four continuously stirred tanks in cascade over a length of 0.87 m. According to Levenspiel [17], the concentration over reactor volume may be approximated using plug-flow reactor behavior. Therefore, the evaluation of the overall reaction rate in respect to $CO_2$ was modelled using an ideal plug-flow reactor approach (Equation (38)) by plotting the overall $CO_2$ conversion X [-] over $V_C/\dot{n}_{CO_2}$. The overall reaction rate is indicated by the slope of the connecting line between 0 and the steady-state operational point, as shown in Figure 20 for a temperature of 25 °C, resulting in reaction rates of 0.17 mol m$^{-3}$ s$^{-1}$ for 1 L min$_{STP}$$^{-1}$ gas flow, 0.36 mol m$^{-3}$ s$^{-1}$ for 2 L min$_{STP}$$^{-1}$ and 0.56 mol m$^{-3}$ s$^{-1}$ for 3 L min$_{STP}$$^{-1}$.

$$V = \dot{n}_{CO_2,0} \cdot \int_0^X \frac{dX}{-r_A} \tag{38}$$

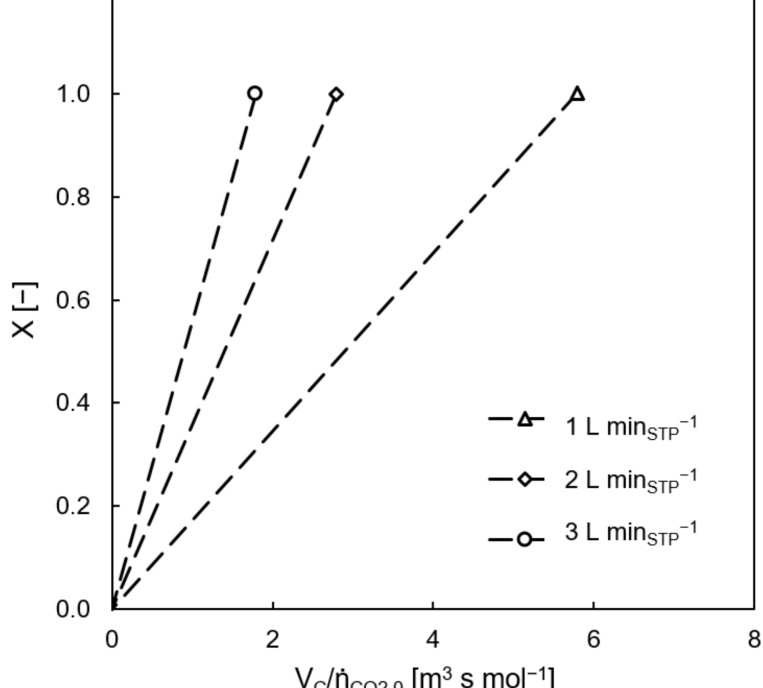

**Figure 20.** Overall $CO_2$ conversion X in the reactor vs. $V_c/n_{CO2}$ at 25 °C. The slope of the dashed lines gives the average reaction rate in the reactor: 1 L min$_{STP}$$^{-1}$ = 0.13 mol m$^{-3}$ s$^{-1}$; 2 L min$_{STP}$$^{-1}$ = 0.36 mol m$^{-3}$ s$^{-1}$; 3 L min$_{STP}$$^{-1}$ = 0.56 mol m$^{-3}$ s$^{-1}$.

It was not possible to investigate the influence of temperature because at these high conversion rates, no significant difference in $CO_2$ consumption was registered for the chosen parameters. Steady-state $CO_2$ concentration at the column outlet is shown in Figure 12. The observed trend in $CO_2$ outlet concentration in a range from 0 to 500 ppm does not affect the $CO_2$ transfer at such a high feed concentration of 30 vol%. The average reaction rate is slightly influenced by the rotational speed. Figure 21 shows the decreasing reaction rate with increasing rotational speed, which agrees well with the decreasing dispersed phase holdup for increasing rotational speed (Figure 13). The reaction rate increases linearly with increasing gas flow rate. It must be mentioned that the feed rate of 0.1 molar NaOH solution also increased with the gas flow rate, as shown in Table 6.

The experimental data from the neutralization experiments at steady-state operation (Section 3.1.4) and hydrodynamic characterization (Section 3.2) were used to model the TCDC as a countercurrent CSTR cascade. The algorithm used for the evaluation of the reaction rate $-r_A$ is explained in Section 2.5. The gas-side mass transfer resistance is not concerned in order of high gas flow rate and intense mixing. The volume-based liquid-side mass transfer coefficient was measured for oxygen in water. According to the film

theory, Equation (39) is used for the calculation of $k_La_{CO_2}$ [34]. The diffusion coefficient $D_{CO_2}$ is calculated by Equation (31). The diffusion coefficient for oxygen $D_{O2}$ at 25°C is $2.22 \cdot 10^{-9}$ m$^2$ s$^{-1}$ [35].

$$\frac{k_La_{O_2}}{k_La_{CO_2}} = \frac{D_{O_2}}{D_{CO_2}} \tag{39}$$

The model principle is shown in Figure 22. The number of continuously stirred tanks is obtained from residence time distribution measurements (Figure 15c). The neutralization process was modelled exemplarily for the neutralization experiment at 25 °C with a gas flow rate of 3 L min$_{STP}$$^{-1}$ of 30 vol% $CO_2$ and 70 vol% $N_2$. The corresponding feed rate of 0.1 molar NaOH was 0.42 L min$^{-1}$. The pH value at the column outlet was pH 9.05. Because of the double jacket, it was not possible to measure the pH trajectory over the column height. Therefore, the pH trajectory was iteratively solved for the reactor model. The evaluation of the pH over column height could be significantly improved by using 2-tracer laser-induced fluorescence (2T-LIF), as suggested by Hlawitschka et al. [10] for non-invasive pH measurement in reactive bubble columns.

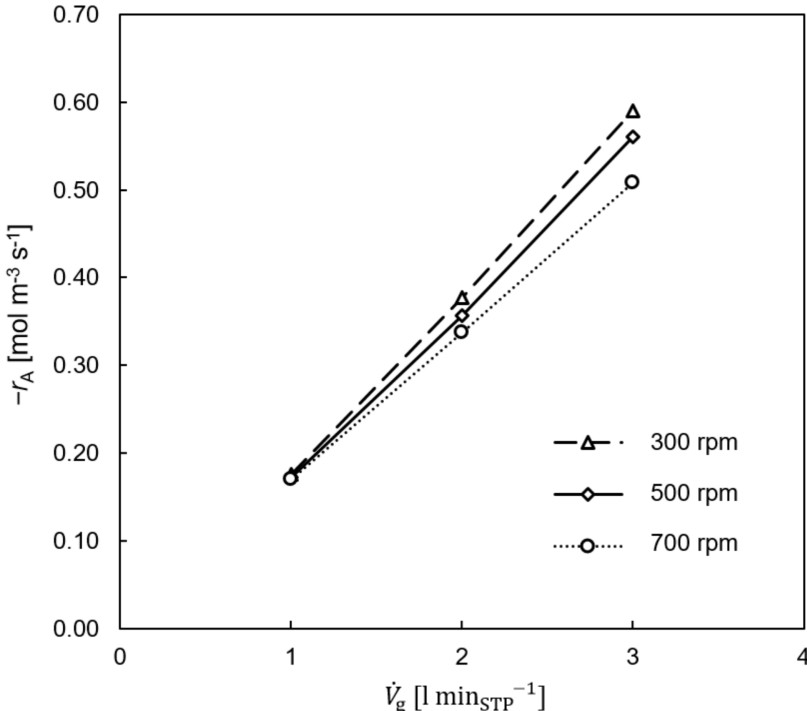

**Figure 21.** Influence of rotational speed and gas flow rate on the reaction rate. Table 6 shows the corresponding aqueous phase feed to the gas flow rate (30 vol% $CO_2$ and 70 vol% $N_2$).

The modelled molar $CO_2$ flux over column height is shown in Figure 23 for five continuously stirred tanks in cascade (dashed line). The dots indicate the molar $CO_2$ flow according to the pH in the individual CSTR, as calculated using the CSTR design equation. For reactor modelling, the volume-specific mass transfer area a [m$^2$ m$^{-3}$] is calculated from the Sauter mean diameter $d_{32}$ and dispersed gas holdup (Equation (40)). It was found that a Sauter mean diameter of 9 mm can represent the $CO_2$ chemisorption in the CSTR cascade approach. This outcome agrees well with visual investigation of slow-motion videos recorded by a high-speed camera. Detailed investigation of bubble size distribution would contribute to the modelling approach.

$$a = \frac{6 \cdot \varphi}{d_{32}} \tag{40}$$

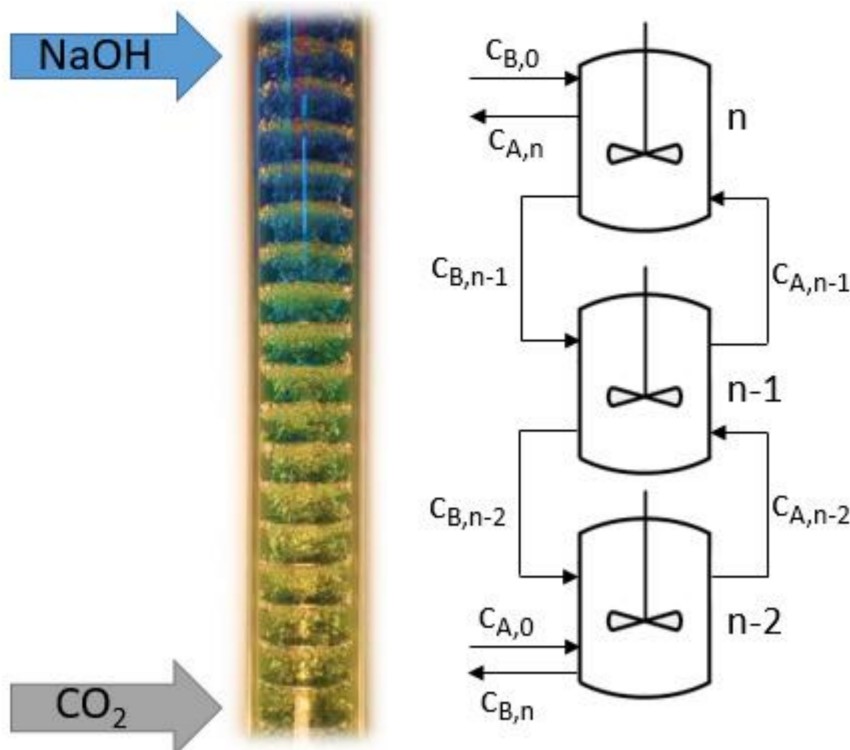

**Figure 22.** Model assumption for continuous chemisorption of $CO_2$ in 0.1 molar NaOH solution in the Taylor-Couette disc contactor. The neutralization process over reactor height was visualized by adding a liquid universal pH indicator. Blue indicates pH 14 at the column inlet and yellow indicates a steady-state pH of pH 9 at the column outlet.

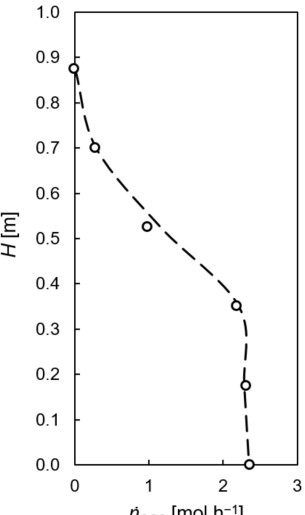

**Figure 23.** Modelled $CO_2$ molar flow rate (dashed line) over column height at steady-state operation for the experimental parameters of 25 °C, 500 rpm, 3 L min$_{STP}^{-1}$ gas feed (30 vol% $CO_2$ and 70 vol% $N_2$), 0.42 L min$^{-1}$ of 0.1 molar NaOH feed with 5 measured CSTRs in a cascade. The dots indicate the molar $CO_2$ flow rate obtained in the individual CSTRs according to the pH with a pKa of 10.14 and 5 CSTRS along the active column height.

The result of the iterative solution for the pH trajectory over column height is shown in Figure 24a. A major pH change is located halfway up the TCDC column. The enhancement factor E was evaluated in every single CSTR using Equation (32). Optical view by pH color

indicator (Figure 22) agrees with these results. The pH-specific ion species were calculated using the Henderson–Hasselbach equation (Equation (4)) and are shown in Figure 24b.

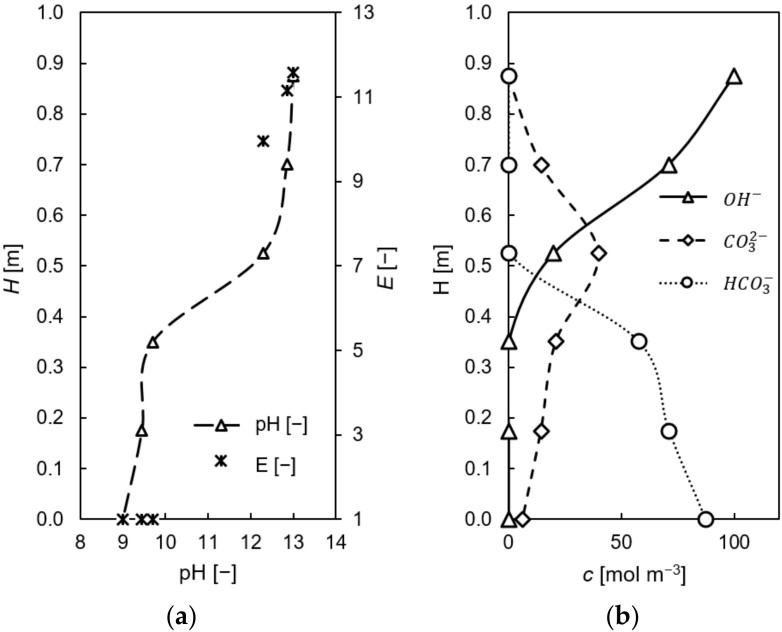

(a)                                    (b)

**Figure 24.** (**a**) Modelled pH trajectory over column height for the neutralization experiment at 25 °C, 500 rpm and gas flow rate of 3 L $min_{STP}^{-1}$ and corresponding enhancement factor E [-] calculated for each CSTR; (**b**) corresponding concentration of ion species with pKa 10.14. The markers indicate the concentration and pH in the individual CSTR.

## 4. Conclusions

The hydraulic performance of continuous countercurrent gas/liquid flow in a Taylor Couette disc contactor was investigated using a non-reactive air-water test system. The rotor disc design for gas/liquid flow was adjusted to the needs of gas/liquid flow by installing perforated rotor discs. Due to the increasing centrifugal force over increasing rotational speed, the volumetric gas phase holdup decreases with increasing rotational speed. Dispersed gas phase holdup between 2.6% at 700 rpm and 1 L $min_{STP}^{-1}$ air and 17.4% at 300 rpm and 3 L $min_{STP}^{-1}$ air was obtained. The residence time distribution of continuous countercurrent flow was modelled using a CSTR cascade model. With the active reactor height of 0.87 m and 24 compartments $N_{CSTR}$ of four to six continuously stirred tanks in series, depending on the flow rate and rotational speed, was deduced from residence time distribution measurements. It was observed that the number of continuously stirred tanks in the cascade increases with increasing rotational speed. This can be explained by the decreasing dispersed phase holdup at higher rotational speed. Characterization of gas/liquid mass transfer was performed by physical absorption of oxygen at steady-state operation. The observed $k_La$ values are in the range between 0.05 $s^{-1}$ at 25 °C and 1 L $min_{STP}^{-1}$ air and 0.2 $s^{-1}$ at 60 °C and 3 L $min_{STP}^{-1}$ of air. The measured hydraulic data were used to model the chemisorption of $CO_2$ in 0.1 molar sodium hydroxide solution. The outlet pH at steady-state operation was set to nine by NaOH feed adjustment. A gas mixture of 30 vol% $CO_2$ and 70 vol% $N_2$ was used for neutralization. The outlet $CO_2$ concentration on top of the column and the pH at the bottom of the column were continuously monitored. $CO_2$ conversions in the reactor were found to be almost 100% for the chosen experimental parameters. The neutralization process was modelled according to the film theory, considsering the reactor height as a cascade of ideal continuously stirred tanks. This study presents the characterization of continuous countercurrent gas/liquid contact in a multiphase Taylor-Couette disc contactor with continuous chemisorption of $CO_2$ in 0.1 molar sodium hydroxide solution. The outcome of the study demonstrates the

successful implementation of a continuous gas/liquid reaction system in the column-type Taylor-Couette disc contactor.

**Supplementary Materials:** The following supporting information can be downloaded at: https://www.mdpi.com/article/10.3390/pr11061614/s1, Video S1: Flow regime 300 rpm, Video S2: Flow regime 500 rpm, Video S3: 700 rpm.

**Author Contributions:** Conceptualization, G.R. and A.G.; methodology, G.R. and R.G.; software, M.V. and G.R.; validation, G.R., R.G. and A.G.; formal analysis, G.R. and R.G.; investigation, G.R., R.G. and M.V.; resources, M.S. and S.L.; writing—original draft preparation, G.R.; writing—review and editing, G.R., M.S., A.G. and S.L.; visualization, G.R. and R.G.; supervision, M.S. and S.L.; project administration, G.R.; funding acquisition, M.S. and S.L. All authors have read and agreed to the published version of the manuscript.

**Funding:** This research received no external funding.

**Data Availability Statement:** Not applicable.

**Acknowledgments:** The authors wish to give their thanks to Alexandra Hutter and Mario Liegl for experimental support. Supported by TU Graz Open Access Publishing Fund. Open Access Funding by the Graz University of Technology.

**Conflicts of Interest:** The authors declare no conflict of interest. The funders had no role in the design of the study; in the collection, analyses, or interpretation of data; in the writing of the manuscript; or in the decision to publish the results.

## Abbreviations

| | | |
|---|---|---|
| $-r_A$ | [mol m$^{-3}$ s$^{-1}$] | reaction rate |
| $\Delta P$ | [Pa] | differential pressure |
| a | [m$^2$ m$^{-3}$] | volumetric interfacial area |
| $A_{ring}$ | [m$^3$] | free cross-sectional are |
| Bo | [-] | Bodenstein number |
| c | [mol m$^{-3}$] | concentration |
| D | [m$^2$ s$^{-1}$] | diffusion coefficient |
| $D_{ax}$ | [m$^2$ s$^{-1}$] | axial dispersion coefficient |
| $d_c$ | [m] | column diameter |
| $d_R$ | [m] | rotor disc diameter |
| $d_{Sh}$ | [m] | column shaft diameter |
| E | [-] | Enhancement factor |
| $E_i$ | [s$^{-1}$] | exit age function |
| $E_{\theta i}$ | [-] | dimensionless exit age function |
| g | [m s$^{-2}$] | earth gravity constant ($g = 9.81$ m s$^{-2}$) |
| $g_i$ | [-] | relative conductivity |
| H | [m] | active reactor length |
| h | [m] | height |
| Ha | [-] | Hatta number |
| $H_c$ | [m] | compartment height |
| He | [mol m$^{-3}$ Pa$^{-1}$] | Henry constant |
| I | [mol m$^{-3}$] | ionic strength |
| k | [m$^3$ mol$^{-1}$ s$^{-1}$] | reaction rate constant |
| $k_l$ | [m s$^{-1}$] | mass transfer coefficient |
| $k_L a$ | [s$^{-1}$] | volumetric mass transfer coefficient |
| N | [-] | number of compartments/ number of CSTRs |
| $\dot{n}$ | [mol s$^{-1}$] | molar flow |
| P | [Pa] | pressure |
| p | [Pa] | partial pressure |
| $P_s$ | [Pa] | saturation pressure |
| r | [m] | radius |
| rpm | [min$^{-1}$] | rotational speed |

| | | |
|---|---|---|
| t | [s] or [°C] | time or temperature; definition is given in the text |
| t | [s] | mean residence time |
| T | [K] or [°C] | tempertaure in °C or K as given in the text |
| u | [m s$^{-1}$] | superficial phase velocity |
| V | [m$^3$] | Volume |
| $\dot{V}$ | [m$^3$ s$^{-1}$] | volume flow |
| V$_C$ | [m$^3$] | column volume |
| **Greek letters** | | |
| $\alpha$ | [-] | Coefficient |
| $\theta_i$ | [-] | dimensionless time |
| $\nu$ | [m$^2$ s$^{-1}$] | kinematic viscosity |
| $\rho$ | [kg m$^{-3}$] | Density |
| $\sigma$ | [-] | variance |
| $\varphi$ | [-] | dispersed gas phase holdup |
| **Subscripts** | | |
| * | | equilibrium |
| g | | gas phase |
| l | | liquid phase |
| w | | water |
| TCDC | | Taylor-Couette Disc Contactor |
| CO$_2$ | | carbon dioxide |
| CSTR | | continuously stirred tank reactor |
| N$_2$ | | nitrogen |
| NaOH | | sodium hydroxide |
| ppm | | parts per million |
| STP | | standard temperature |

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
