# Peer review of "Gas/Liquid Operations in the Taylor-Couette Disc Contactor: Continuous Chemisorption of CO2"

_processes, doi:10.3390/pr11061614_

Round 1

Reviewer 1 Report

Please see the uploaded file

Author Response

Changes in the manuscript are highlighted in the change track mode.

Reviewer 2 Report

Gas/liquid contactors are commonly used in chemical and biotechnological applications, and their design requires knowledge of appropriate gas flow patterns. This study presents a flexible and easy-to-apply design for a Taylor-Couette Disc Contactor that provides constant gas phase holdup and contact area over reactor height without a separate gas distributor. The presented experimental study investigates the performance of the contactor in gas/liquid mass transfer operations, including continuous neutralization of sodium hydroxide with a gas mixture of CO2 and N2, and deduces the volume-based liquid side mass transfer coefficient kLa to model the reaction according to the two film theory over the column height.

The article is written at a high scientific level. It is evident that the authors have put significant effort into reviewing and synthesizing the available literature, and their experimental investigation represents a significant contribution to the field. Detailed description of the experimental study is presented by authors. The article is well written and presents a wealth of information that will be of interest to researchers and postgraduate students alike. There are a few comments and questions to the article that need to be answered by the authors:

1.     In Table 1, the authors presented the dimensions of the TCDC device. It would be nice to provide a TCDC picture where these dimensions will be shown. You can add this information on Figure 3.

2.     At first the authors provide data on dispersed gas phase holdup in section “2.3 Rotor disc design for gas/liquid contact”, and after in section “2.4 Column hydrodynamics” explains what it is and how it is defined (equation 5). In my humble opinion, it should be the opposite, and the results presented in pictures 4 and 5 should be moved to the section " 3. Results".

3.     To characterize the flow regime, the authors mention the Reynolds and Taylor numbers, about the authors do not apply them in the article. RPM is used instead. What is the reason for mentioning Reynolds and Taylor numbers? The title of the article mentions the Taylor-Couette flow, but it is not clear under what flow regimes they appear. Also, same about the superficial phase velocity u and hydraulic load B. It is unclear their use in the article.

4.     To denote hydraulic load and coefficient in equation 22 authors use Letter B. Please add this information to nomenclature (5. Symbols).

5.     Why is ??i defined differently in equations (15) and (37)?

Author Response

Dear Reviewer, 

thank you for the constructive feedback. The answers to your questions can be found in the uploaded file.

Reviewer 3 Report

Dear Author(s) 

The article is in general good and results are supported with data but there are some minor suggestions that I have:

1) The aim of the article is not clear to me in the introduction or in the abstract.  It is mentioned that the work presents flexible and easy-to-apply column design . . .Can you please rephrase or make the aims more clear for the reader, dispersed gas holdup and reactor modelling were done for instance for a purpose.

2) The word "flow" is usually used as a prefix for words in the article

Example: Flowmeter, flowrate where it should be flow meter, flow rate etc.

3) Figure 11: I don't quite understand the caption in Figure 11, please rephrase.  Can you explain why the error bar range for pH is higher at 40 °C than at 25 °C and 60 °C?  You mention that error is due to the effect of rotational speed but why is the error range widest/highest at 40°C?

The English language in the article is generally reasonable.

Author Response

(The authors gave the same response as above.)

Round 2

Reviewer 2 Report

The authors did a good job on the article, made changes and improvements, and gave fully comprehensive answers to the questions. The article is ready for publication.